# Indocyanine Green-Nexturastat A-PLGA Nanoparticles Combine Photothermal and Epigenetic Therapy for Melanoma

**DOI:** 10.3390/nano10010161

**Published:** 2020-01-17

**Authors:** Debbie K. Ledezma, Preethi B. Balakrishnan, Juliana Cano-Mejia, Elizabeth E. Sweeney, Melissa Hadley, Catherine M. Bollard, Alejandro Villagra, Rohan Fernandes

**Affiliations:** 1The Institute for Biomedical Sciences, The George Washington University, 2300 Eye Street NW, Room 561, Washington, DC 20037, USA; dkledezma@gwmail.gwu.edu (D.K.L.); cbollard@childrensnational.org (C.M.B.); 2The George Washington Cancer Center, The George Washington University, 800 22nd St NW, Science and Engineering Hall, Suite 8000, Washington, DC 20052, USA; pbalakrishnan@gwu.edu (P.B.B.); jcm22@terpmail.umd.edu (J.C.-M.); lizie@gwu.edu (E.E.S.); hadleym@gwu.edu (M.H.); avillagra@gwu.edu (A.V.); 3Center for Cancer and Immunology Research, Children’s National Health System, 111 Michigan Ave NW, Washington, DC 20010, USA; 4Department of Medicine, The George Washington University, 2150 Pennsylvania Avenue NW, Suite 8-416, Washington, DC 20037, USA

**Keywords:** PLGA nanoparticles, photothermal therapy, epigenetic therapy, melanoma, indocyanine green, HDAC inhibitors, Nexturastat A

## Abstract

In this study, we describe poly (lactic-*co*-glycolic) acid (PLGA)-based nanoparticles that combine photothermal therapy (PTT) with epigenetic therapy for melanoma. Specifically, we co-encapsulated indocyanine green (ICG), a PTT agent, and Nexturastat A (NextA), an epigenetic drug within PLGA nanoparticles (ICG-NextA-PLGA; INAPs). We hypothesized that combining PTT with epigenetic therapy elicits favorable cytotoxic and immunomodulatory responses that result in improved survival in melanoma-bearing mice. We utilized a nanoemulsion synthesis scheme to co-encapsulate ICG and NextA within stable and monodispersed INAPs. The INAPs exhibited concentration-dependent and near-infrared (NIR) laser power-dependent photothermal heating characteristics, and functioned as effective single-use agents for PTT of melanoma cells in vitro. The INAPs functioned as effective epigenetic therapy agents by inhibiting the expression of pan-histone deacetylase (HDAC) and HDAC6-specific activity in melanoma cells in vitro. When used for both PTT and epigenetic therapy in vitro, the INAPs increased the expression of co-stimulatory molecules and major histocompatibility complex (MHC) Class I in melanoma cells relative to controls. These advantages persisted in vivo in a syngeneic murine model of melanoma, where the combination therapy slowed tumor progression and improved median survival. These findings demonstrate the potential of INAPs as agents of PTT and epigenetic therapy for melanoma.

## 1. Introduction

Melanoma is a prevalent cancer of the skin, accounting for over 7000 deaths in the United States in 2019 [1]. While the 5-year relative survival is 98.7% for localized melanoma, the survival for patients with regional and metastatic melanoma is dismal (64.7% and 24.8%, respectively). Hence, there is an urgent need to develop novel therapies for this patient population. Since melanoma presents on the skin, it is readily accessible for localized interventions that can elicit potent systemic antitumor immune responses, which can improve the prognosis for patients with regional and metastatic disease. Nanoparticles serve as a suitable platform to develop such localized interventions since they have been used to package and deliver diverse therapeutic agents. In particular, the polymer poly (lactic-co-glycolic acid) (PLGA), which has excellent biocompatibility and biodegradability, has been U.S. Food and Drug Administration (FDA) approved and extensively used to deliver synergistic drug combinations [2,3,4,5]. These properties make PLGA an excellent candidate for applications involving the controlled release of encapsulated agents and improved drug pharmacokinetics in vivo [6,7]. Several published studies have demonstrated that loading drugs within PLGA can enhance their functionality over free drugs by improving their pharmacokinetic profiles and decreasing drug thresholds [4,7,8,9]. For melanoma, recent preclinical studies highlight PLGA as a suitable carrier for immune-stimulating molecules and anti-angiogenic agents [10,11,12]. Additionally, PLGA nanoparticles have also been synthesized with improved colloidal stability and biofunctional coatings that could help improve their use for therapy [13,14]. An emerging area of investigation with PLGA nanoparticles is their use in the field of immunoengineering, wherein agents packaged within the nanoparticles elicit responses from the immune system. To this end, PLGA nanoparticles have been utilized to package and deliver antigens, immune adjuvants, and immunostimulatory drugs to elicit tumor-specific responses and stimulate immune cell activation to enhance cancer immunotherapy [15,16,17].

Premised on these studies, we utilized PLGA nanoparticles to co-encapsulate and administer photothermal therapy (PTT) and epigenetic therapy as a novel combination therapy for melanoma. We encapsulated indocyanine green (ICG), a US FDA-approved photoactive dye as the PTT agent, and Nexturastat A (NextA), an epigenetic drug, within PLGA nanoparticles, generating ICG-NextA-PLGA nanoparticles (INAPs) to co-administer PTT and epigenetic therapy. PTT is an effective method for localized tumor ablation using light-responsive nanoparticles and a wavelength-matched light source. Typically, near infrared (NIR) light-responsive nanoparticles, including nanoparticles containing ICG, are illuminated with an NIR laser generating heat, which triggers tumor cell death [18]. Previous studies have demonstrated that PTT can increase tumor immunogenicity, elicit immunogenic cell death, and be combined with immunological adjuvants (e.g., toll-like receptor agonists) or immunotherapies (e.g., checkpoint inhibitors) to treat cancer [19,20,21,22]. However, PTT alone is not sufficient to generate effective systemic antitumor immunity. To complement the effects of PTT, we co-encapsulated the epigenetic drug NextA, a histone deacetylase (HDAC) 6 inhibitor, within the PLGA nanoparticle.

In preclinical studies with melanoma murine models, HDAC6 inhibitors exhibited antitumor activity and immune marker modulation to increase melanoma immunogenicity through major histocompatibility complex Class I (MHC-I) and tumor antigen expression on melanoma cells [23,24,25]. Other groups have also demonstrated that HDAC6 promotes melanoma proliferation and migration, suggesting its potential as a therapeutic target to control tumor growth [26,27]. While NextA has been shown to specifically inhibit HDAC6 activity, these effects have not been sufficient to generate complete tumor regression in melanoma, which can be achieved using PTT. Regardless, NextA differs from other HDAC inhibitors that are currently FDA approved for epigenetic therapy. These approved HDAC inhibitors, including vorinostat, romidepsin, and panobinostat, are effective monotherapies for hematological malignancies; however, they exhibit limited efficacy in treating solid tumors, as evidenced by low response rates and severe side effects in patients [28,29,30,31,32]. Poor pharmacokinetics and nonspecific HDAC inhibition by these drugs are potential reasons for this limited efficacy. By packaging the HDAC6 inhibitor NextA along with ICG into PLGA nanoparticles, we aimed to leverage the advantages of NextA for melanoma while mitigating some of the aforementioned limitations observed with PTT and epigenetic therapy as single therapies. 

We hypothesized that the combination of PTT with epigenetic therapy administered using the INAPs would promote better tumor immunogenicity after PTT ablation, leading to slower tumor progression and improved survival in tumor-bearing mice (Figure 1). Compared with earlier studies, our approach using a single PLGA nanoparticle-based platform to combine PTT and epigenetic therapy is novel. The specific components, ICG, NextA, and PLGA, that constitute the INAPs are all US FDA approved, providing a sound rationale for combining these non-overlapping yet complementary therapies. Additionally, the INAPs are administered locally, and based on previous degradation studies with PLGA-based nanoparticles, we expected the INAPs to completely bio-degrade, which mitigates concerns associated with the fate and toxicity of the nanoparticles post-treatment [6,33,34,35]. Finally, to our knowledge, the specific combination of ICG-based PTT with NextA-based epigenetic therapy for melanoma using PLGA nanoparticles has not been previously described. In this work, we demonstrate a proof-of-concept approach using a PLGA nanoparticle-based platform to combine PTT and epigenetic therapy. We present a facile nanoemulsion synthesis scheme to synthesize the INAPs. We characterized the size, stability, and encapsulation efficiency of the INAPs. Studies assessing both the PTT and epigenetic therapy capabilities of the INAPs are presented. Next, we tested the ability of the INAPs to modify the expression of immunomodulatory markers on melanoma cells in vitro. Finally, we assessed the efficacy of the INAPs in treating melanoma in a syngeneic, SM1 murine model of melanoma.

## 2. Materials and Methods 

### 2.1. Chemical and Biological Reagents

Poly (lactic-co-glycolic acid) (PLGA; lactide: glycolide (50:50) and MW 30–60 kDa), poly (vinyl alcohol) (PVA; MW 89–98 kDa and 99+% hydrolyzed), acetonitrile, indocyanine green (ICG), and dimethyl sulfoxide (DMSO) were purchased from Sigma-Aldrich (St. Louis, MO, USA). Nexturastat A (NextA) was obtained from StarWise Pharmaceuticals (Madison, WI, USA). The Cell Titer-Glo™ viability assay and the HDAC-Glo™ I/II Assay and Screening System kits were purchased from Promega (Madison, WI, USA). Phosphate-buffered saline (PBS) was purchased from Thermo Fisher Scientific (Waltham, MA, USA). The immunoblot antibodies, anti-acetylated-α-tubulin rabbit (3971S), anti-α-tubulin mouse (3873S), anti-acetylated-H3 rabbit (9649S), and anti-H3 mouse (3638S), were all purchased from Cell Signaling (Danvers, MA, USA). The antibodies used for flow cytometry, Alexa Fluor 647 anti-mouse CD80 (clone 16-10A1), PE anti-mouse CD86 (clone GL-1), and Alexa Fluor 488 anti-mouse H-2k^b^ (clone AF6-88.5), were all purchased from BioLegend (San Diego, CA, USA). The water used in all studies was obtained from a Milli-Q ultrapure water system (Millipore Corporation, Billerica, MA, USA) with a resistivity of 18.2 MΩ-cm.

### 2.2. Cells Lines and Cell Culture

The murine melanoma cell line SM1 (a BRAF^V600E^-driven melanoma) was obtained from the laboratory of Dr. A. Ribas at the University of California Los Angeles [36]. The murine melanoma cell line B16F10 was purchased from ATCC (Manassas, VA, USA). Cell culture media Dulbecco’s modified Eagle Medium (DMEM) was purchased from Thermo Fisher Scientific and Hyclone RPMI 1640 media was purchased from GE Healthcare Life Sciences (Pittsburgh, PA, USA). When needed, cell culture media was supplemented with 1% penicillin/streptomycin (PenStrep) purchased from Corning (Corning, NY, USA). Fetal bovine serum (FBS) was obtained from Thermo Fisher Scientific.

### 2.3. Animals and Animal Studies

All animal studies were conducted in accordance with protocols approved by the Institutional Animal Care and Use Committee of the George Washington University (Protocols A396 and A354). The studies were conducted to ensure humane care of the animals as per the IACUC’s guidelines. Further, 4- to 6-week-old female C57B/6 were purchased from Jackson Laboratory (Farmington, CT, USA). Prior to initiating any study, the animals were acclimated for 7 days.

### 2.4. Synthesis of ICG-NextA-PLGA Nanoparticles (INAPs)

INAPs were synthesized using a nanoemulsion synthesis scheme. Briefly, NextA (10 mg/mL) and ICG (10 mg/mL) were both dissolved in DMSO, and 50 µL of each were added to 1 mL of solution containing PLGA (20 mg/mL) in acetonitrile, and mixed thoroughly by vortexing. This organic phase was then added to 5 mL of aqueous solution containing 5% PVA, and then vortexed for approximately 30 s to generate a homogeneous emulsion. The emulsion was then transferred to a beaker and stirred at 400 rpm for 3 h at room temperature for solvent evaporation. The resulting INAPs were collected by centrifugation at 10,000× *g* for 30 min and resuspended in Milli-Q water by two sonication cycles (40% amplitude at 30 s per cycle) on an ice bath. As controls, ICG-PLGA nanoparticles (ICG-PLGA), NextA-PLGA nanoparticles (NextA-PLGA), and blank-PLGA nanoparticles (Blank-PLGA) were synthesized using an identical synthesis scheme by adding 50 µL of either ICG (for ICG-PLGA) or NextA (for NextA-PLGA) with 50 µL of DMSO or 100 µL of DMSO (for Blank-PLGA) in the organic phase.

### 2.5. Size Characterization

The hydrodynamic sizes of the nanoparticles were determined using a Zetasizer Nano ZS (Malvern Instruments, Worcestershire, UK). The dispersant used for all measurements for the dynamic light scattering (DLS) analysis was deionized water (Millipore Sigma, Burlington, MA, USA) with a typical concentration of 10 µg/mL nanoparticles. All readings were done in triplicate with at least 15 scans per replicate using a 633 nm laser and a 173° detection angle. 

### 2.6. SEM Characterization of PLGA Nanoparticles

The morphology and structure of PLGA nanoparticles were characterized by scanning electron microscopy (SEM) using FEI Teneo LV FEG SEM (Thermo Fisher Scientific) with the Everhart-Thornley Detector (ETD) for secondary and back-scattered electrons. All types of PLGA nanoparticles were visualized using a voltage (HV) set to 2.00 kV, and beam current (curr) set to 13, 25, or 50 pA depending on the magnification. The magnification varied with each image (refer to figure caption for this detail). The obtained images using the ETD had an electron beam dwell time of 10 microseconds and resolution of 1536 × 1026 px. 

### 2.7. Encapsulation Efficiency

After synthesis of the PLGA nanoparticles, the concentrations of the INAPs and control nanoparticles (mg/mL) used in the studies were determined by oven drying a fixed volume of the nanoparticles at 80 °C for 1 h and measuring the resultant nanoparticle mass. To quantify the drug loading efficiency (for both ICG and NextA), a fixed mass of dried nanoparticles was dissolved in DMSO and the concentrations of the loaded drugs were determined by ultraviolet-visible-near infrared (UV-Vis-NIR) spectroscopy using a Nanodrop (Thermo Fisher Scientific). Standard curves for free ICG and free NextA were used to determine the encapsulation efficiency. The encapsulation efficiency for NextA in INAPs and NextA-PLGA was determined by assessing their UV-Vis spectra and then blanking with the spectra of ICG-PLGA and Blank-PLGA to calculate the contribution of NextA. The encapsulation efficiency for ICG was determined similarly for INAPs and ICG-PLGA by blanking the spectra of Blank-PLGA. The encapsulation efficiency was calculated as the amount of drug encapsulated expressed as a percentage of the amount of drug utilized in the synthesis. 

### 2.8. Photothermal Properties of INAPs 

The photothermal heating profiles for INAPs were determined as a function of the nanoparticle concentration (0.5–4.0 mg/mL) by diluting the nanoparticles in media. Nanoparticles were irradiated with an 808 nm near infrared (NIR) laser for 5 min (Laserglow Technologies, Toronto, ON, Canada). The photothermal properties of the INAPs (4 mg/mL) were also measured at varying powers (0.6–1.2 W) for 5 min. The laser power was confirmed using a power meter (Thorlabs, Newton, NJ, USA). Temperatures were recorded each minute using an infrared thermal camera (FLIR, Arlington, VA, USA). The thermal dose was calculated using cumulative equivalent minutes at 43 °C (CEM43), as previously described [37]. 

### 2.9. Cellular Viability of Melanoma Cells 

The viability of INAPs-treated SM1 or B16F10 (1 × 10^6^ cells) was determined at varying nanoparticle concentrations (0.5–2.0 mg/mL) in the presence or absence of an NIR laser by suspending cells in PBS (200 µL). Post-PTT, the cells were then transferred to 6-well plates and incubated in media for 24 h. The cells were then collected, suspended in 400 µL of PBS, and assessed for viability in triplicate using the Cell Titer-Glo ATP assay (following the manufacturer’s instructions) (Promega). As controls, ICG-PLGA at 0.5 to 2.0 mg/mL and free NextA (5 µM) were used. Luminescence was measured using a SpectraMax i3x Multi-mode microplate reader from Molecular Devices, LLC (San Jose, CA, USA). 

### 2.10. HDAC Activity Assay

The functionality of encapsulated NextA was determined using an HDAC-Glo™ I/II assay and screening system kit. Melanoma cells seeded at 10,000 cells per well in a white 96-well flat-bottom plate were incubated overnight at 37 °C and treated with INAPs (1.25 mg/mL) for 1 h. INAPs were diluted in media (4.0 mg/mL) and either incubated at 37 °C or irradiated with the NIR laser before treating cells. The HDAC-Glo™ I/II assay HDAC-Glo media mix was added to each well under minimal light exposure. Luminescence was immediately measured afterwards at a signal-steady kinetic state using the SpectraMax plate reader. Each treatment was repeated in triplicate. Cells were treated with panobinostat (LBH) (2.5 µM) as a positive control. Blank-PLGA, NextA-PLGA, ICG-PLGA, DMSO, and Milli-Q water were used as controls. 

### 2.11. Immunoblotting 

SM1 cells were cultured with complete RPMI media (RPMI, 10% FBS, 1% PenStrep) and plated at a 250,000 cell density in 6-well plates overnight. Cells were treated with nanoparticles (1 mg/mL) for 24 h and then harvested with RIPA buffer containing protease and phosphatase inhibitors, purchased from Thermo Fisher Scientific. Cell lysates were sonicated for 8 min (8 cycles of 30 s on/off) using a water bath sonicator purchased from Diagenode (Denville, NJ, USA) and then centrifuged in a microcentrifuge at 16,000× *g* at 4 °C. Protein concentrations in the supernatants were measured with the Pierce BCA protein assay purchased from Thermo Fisher Scientific, according to the manufacturer’s instructions. Proteins from the cell lysate were prepared for immunoblotting by diluting 10 µg of protein in 4× sample buffer and 10× reducing agent from Thermo Fisher Scientific. Samples were run on a 4–20% BioRad gel and transferred onto a polyvinylidene difluoride (PDVF) membrane using a transfer system from BioRad (Hercules, CA, USA). Membranes were blocked with a 1:1 Odyssey buffer in PBS. Protein bands were detected with an Azure imaging system. 

### 2.12. Immunomodulation of Melanoma Cells in Vitro

In these studies, the melanoma cells were resuspended in PBS (200 µL) and treated with 2.0 mg/mL of INAPs or ICG-PLGA in the presence or absence of an NIR laser at 0.4 W for 5 min. Other controls included untreated, vehicle (water), vehicle (DMSO), and free NextA (5 µM). After 24 h of incubation post-treatment, the supernatants and cells were collected and rinsed with PBS at least twice. The cells were then resuspended in 200 µL of PBS containing 1 µL of Zombie Violet^TM^/Green^TM^/Aqua^TM^ fixable viability kit and incubated at 4 °C for 20 min in the dark. Next, the cells were rinsed with PBS, collected at 400× *g* for 3 min, and then resuspended in 100 µL of flow buffer (1% FBS in 1 × PBS) containing 5 µL of mouse Fc blocking buffer and incubated for another 10 min. After cells were washed and collected, cells were resuspended in flow buffer containing flow antibodies (1 µL PE-CD86, 1 µL AlexaFluor647-CD80, 1 µL AlexaFluor 488-H-2kb) in 100 µL, and incubated for 1 h. Cells were then washed and resuspended in 200 µL of flow buffer and then stored at 4 °C overnight. Samples were run on a Celesta Cell Analyzer with HTS (BD BioSciences, Franklin Lakes, NJ, USA) and the analysis was conducted on a FlowJo^TM^ v10.6.1. Respective fluorescence minus one (FMO) controls were used for gating analysis. UltraComp beads from Thermo Fisher Scientific were used for single fluorophore controls (compensation controls).

### 2.13. Tumor Growth and Survival

Syngeneic melanoma tumor-bearing mouse models were established by subcutaneously injecting 1 million SM1 melanoma cells onto the shaved backs of female C57BL/6 mice. Tumor growth was monitored by measuring the length and width of the tumor using calipers. For in vivo PTT, SM1 tumor-bearing mice with tumors of at least 60 mm^3^ in size were anesthetized with 2% to 5% isoflurane and treated with the INAPs (50 mg/kg) via intratumoral (i.t.) injection. Post-injection, the tumor area was immediately irradiated with the NIR laser with the NIR laser power adjusted (maximum reached was 0.4 W) to maintain a temperature of 90 to 100°C for 10 min. Mice were treated according to the following groups: (1) Untreated—receiving no treatment; (2) free NextA (intraperitoneal (i.p.), 25 mg/kg)—receiving free NextA (epigenetic therapy) 6× per week; (3) INAPs (i.t., 50 mg/kg)—receiving i.t. encapsulated NextA (epigenetic therapy); (4) ICG-PLGA-PTT (i.t., 50 mg/kg + NIR laser for 10 min)—receiving PTT; (5) INAPs-PTT (i.t., 50 mg/kg + NIR laser for 10 min)—receiving both PTT and epigenetic therapy; and (6) INAPs-PTT (i.t. 50 mg/kg + NIR laser for 10 min) + 2 NextA-PLGA boosters (i.t. 50 mg/kg)—receiving both PTT and epigenetic therapy. For the free NextA group, NextA was administered daily by intraperitoneal injection until the tumor exceeded 20 mm in any dimension. When used, NextA-PLGA boosters (50 mg/kg) were i.t. administered on day 3 and 7 after PTT. 

### 2.14. Animal Exclusion and Euthanasia Criteria

Animals were excluded from the study if their tumors failed to grow after SM1 inoculation. This exclusion criterion was not used in this study as all injected mice developed tumors. Mice were euthanized when tumor sizes exceeded 20 mm in any dimension. Euthanasia was achieved through cervical dislocation after CO_2_ narcosis. If the tumor impaired the mobility of the animal, became ulcerated or appeared to be infected, or if the mice displayed any signs of distress, such as assuming a sick mouse posture, the mice were immediately excluded from the study and euthanized. All these steps were conducted in accordance with the approved Institutional Animal Care and Use Committee (IACUC) protocols. 

### 2.15. Statistical Methods for this Study

Statistical significance was determined using one-way analysis of variance (ANOVA), two-way ANOVA, and Tukey and Sidek’s multiple comparisons tests. Values with *p* < 0.05 were considered statistically significant. Animal survival across the various treatment and control groups were analyzed by generating Kaplan–Meier curves.

## 3. Results 

### 3.1. INAPs Successfully Co-Encapsulate ICG and NextA within PLGA Nanoparticles

To co-administer PTT and epigenetic therapy using a single nanoparticle, ICG (a photoactive dye) and NextA (an epigenetic drug) were encapsulated within PLGA nanoparticles using a nanoemulsion synthesis scheme. The synthesis resulted in monodispersed INAPs with a mean hydrodynamic diameter of 220 nm and polydispersity index of 0.103 (Figure 2a). Control nanoparticles containing ICG (ICG-PLGA), NextA (NextA-PLGA), or vehicle (Blank-PLGA) exhibited similar monodisperse size distributions (Figure 2a). Importantly, the INAPs exhibited multi-day stability (over 87 days) when suspended in Milli-Q water, as evidenced by their retained uniform size distributions (Figure 2b). SEM images of the INAPs confirmed their uniform spherical morphology (Figure 2c). There were minimal variations in the morphology of INAPs when compared to Blank-PLGA (Appendix A). INAPs retained the characteristic absorption band of ICG at 780 nm (Figure 2d) and that of NextA at 250 nm (Figure 2e) as measured by UV-Vis-NIR spectrometry, illustrating the encapsulation of both agents. For both free NextA and free ICG, the level of absorbance increased with the increasing concentration (Appendix A). The encapsulation efficiency of the synthesized INAPs was typically 40% to 50% for ICG and 30% to 70% for NextA, and drug loading was 11 µg ICG/mg INAP and 7 to 18 µg NextA/mg INAP. These findings demonstrate that both the PTT agent ICG and the epigenetic drug NextA can be successfully co-encapsulated within INAPs using the nanoemulsion synthesis scheme. Likewise, stable modular PLGA nanoparticles containing either ICG (ICG-PLGA) or NextA (NextA-PLGA) can be generated using this scheme. 

### 3.2. INAPs Can Be Photothermally Heated and Trigger Cell Death in Melanoma Cells in Vitro

We conducted studies in vitro to assess the photothermal properties of the INAPs. When illuminated with an 808 nm NIR laser, INAPs heated in a concentration-dependent manner, reaching temperatures greater than 50 °C at concentrations ≥2.0 mg/mL and a laser power of 0.8 W (Figure 3a). The concentration-dependent heating is again reflected when the time–temperature heating curves (Figure 3a) were expressed in terms of the cumulative equivalent minutes at 43 °C (CEM43; Figure 3b), a parameter that allows comparison of the thermal doses administered [37]. The INAPs heating was also NIR laser power dependent, with increasing temperatures achieved with increasing laser powers (Figure 3c,d). UV-Vis-NIR spectroscopy demonstrated that the encapsulated ICG degraded after PTT (INAPs-PTT) as evidenced by a decrease in the characteristic ICG absorption peak compared to the control nanoparticles (INAPs), which exhibited an intact ICG absorption peak similar to free ICG (Figure 3e). SEM images also captured the morphology change, wherein the INAPs lost their nanoparticulate structure after PTT, indicating that the INAPs were effective single-use PTT agents (Figure 3f). Interestingly, INAPs suspended in complete media maintained a similar size distribution before and after irradiation with the NIR laser when compared to INAPs suspended in PBS (Appendix A). This can be attributed to the protection conferred by proteins from the serum (FBS) adsorbing onto the surface of nanoparticles, a phenomenon known as the protein corona effect [38]. The time–temperature heating curves reflect this protection, with maximum INAPs protection at 20% FBS compared to INAPs in media only (without FBS; Appendix A). As a final component of this study, we tested the ability of the INAPs to be used for PTT of melanoma cells in vitro. When INAPs-PTT was performed on SM1 cells, cell viability decreased to 20% when heated to ~50°C, which was achieved at a 2 mg/mL INAP concentration (Figure 3g and Appendix A for statistical analysis). Similarly, ICG-PLGA-PTT generated an equivalent decrease in the viability of SM1 cells (also at 2 mg/mL), indicating that the encapsulated NextA had a minimal effect on SM1 cell viability in the context of PTT (Figure 3g). When melanoma cells were treated with 3 or 10 mg/mL INAPs for PTT, we determined that PTT at 0.8 W for 10 min at a concentration of 3 mg/mL or higher reduced tumor viability to 1% to 2%, thus serving as a positive control for administering PTT (Appendix A). Based on the NextA toxicity alone (Appendix A), we observed a decrease in the viability in SM1 cells starting at 11 µM in a NextA dose-dependent manner (to 100 µM). This suggests that if the INAPs alone exhibit high viability, then the functional NextA released from the INAPs is significantly lower than 11 µM. To ensure our epigenetic therapy with NextA elicited only immunomodulation in combination with PTT, we performed subsequent studies with free NextA at 5 µM to minimize any cytotoxicity effects in our combination. An additional rationale for selecting this dose is that previously published studies utilized 5 µM NextA for immunomodulatory effects in melanoma cells [24,25]. Overall, these results demonstrate that the INAPs heat in a concentration- and laser power-dependent manner and can be used for a single administration of PTT, after which they exhibit negligible PTT properties. Further, INAPs-PTT elicit cell death in targeted SM1 melanoma cells in vitro. 

### 3.3. NextA Encapsulated within INAPs Can Inhibit HDAC Activity in Melanoma Cells in Vitro

To determine the functionality of NextA within the INAPs, we conducted studies to assess whether NextA encapsulated within INAPs can inhibit HDAC activity of SM1 and B16F10 melanoma cells in vitro using a luminescence-based HDAC activity assay (measuring pan-HDAC activity). SM1 murine melanoma cells have been used to study immunosensitization for melanoma treatment, whereas B16F10 melanoma cells help study therapeutic applications for metastatic melanoma. We assessed HDAC6 inhibition in both cell lines to verify that NextA could function in both cell types. First, we compared the ability of the INAPs, control nanoparticles (ICG-PLGA, NextA-PLGA, and Blank-PLGA), free drug (NextA), and controls to inhibit HDAC activity in the presence or absence of laser exposure. Importantly, INAPs, NextA-PLGA, free NextA, and the positive control (panobinostat; LBH) all significantly inhibited HDAC activity (Figure 4a). SM1 cells had HDAC activity levels of less than 5% when treated with INAPs, NextA-PLGA, and LBH, and less than 20% when treated with free NextA, as compared to the appropriate controls (vehicle-treated cells for free agents and Blank-PLGA for NextA-containing nanoparticles; Figure 4a). Although cells treated with ICG-PLGA appeared to decrease HDAC activity levels relative to the control, these changes were not statistically significant (please refer to Appendix A for the statistical analysis for this study). Importantly, the addition of the laser did not interfere with the ability of the encapsulated NextA (within INAPs and NextA-PLGA) or free NextA to inhibit HDAC activity as measured by the unchanged HDAC activity levels for the various treatment groups with/without laser exposure (Figure 4a). The INAPs inhibited HDAC activity in a concentration-dependent manner, with increasing INAP concentrations eliciting decreased HDAC activity levels relative to controls (Figure 4b). Once again, the addition of the laser did not interfere with the observed HDAC activity levels for a given INAP concentration. As a third component of this study, we tested whether short-term storage of the INAPs (over 7 days at 4 °C) impacted their ability to inhibit HDAC activity relative to freshly synthesized INAPs. Encouragingly, the INAPs stored for 7 days at 4 °C were able to inhibit HDAC activity at levels similar to freshly synthesized INAPs in the presence/absence of NIR laser exposure (Figure 4c). To assess the HDAC6-specific inhibition capabilities (versus pan-HDAC inhibition) of the INAPs and controls, we measured the expression levels of acetylated alpha-tubulin (Ac-α-tubulin) relative to deacetylated alpha-tubulin (α-tubulin) by western blot. INAPs treatment induced higher expression levels of Ac-α-tubulin relative to control treatments, confirming the HDAC6-specific inhibition via the encapsulated NextA within the INAPs (Figure 4d and Appendix A). However, the Ac-α-tubulin expression levels for INAPs were three-fold lower compared to free NextA (Figure 4d). We cannot discount the fact that the presence of free NextA (in the NextA group) directly in contact with target cells in vitro may have generated higher Ac-α-tubulin expression levels relative to the encapsulated NextA being released from the nanoparticles (in the INAP group). However, since improved in vitro performance does not necessarily translate to superior in vivo performance, particularly in the context of blood circulation and perfusion, we did not regard the increased expression levels of Ac-α-tubulin in the free NextA treatment group compared to the INAP treatment group to pose a problem for additional testing. These results demonstrate that NextA within the INAPs can inhibit both pan-HDAC activity and HDAC6-specific activity. Further, the NextA within the INAPs retains its functionality even after NIR laser exposure and after short-term storage. 

### 3.4. INAPs-PTT Increases the Expression of Co-Stimulatory Molecules CD86 and CD80, and MHC Class I Molecules on Melanoma Cells in Vitro 

Our findings thus far have demonstrated that the INAPs retain their PTT properties as well as their ability to inhibit HDAC6 expression in melanoma cells (SM1 and B16F10) in vitro. Next, we studied the effects of combined photothermal and epigenetic therapy on tumor immunogenicity. 

Previous studies have demonstrated tumor immunomodulation of immune markers, such as MHC-I, in melanoma cells after treatment with HDAC6 inhibitor. For this study, we administered INAPs, ICG-PLGA, the free NextA, and controls with and without NIR laser exposure to SM1 melanoma cells in vitro. We determined the effects of the treatments on the surface expression of co-stimulatory molecules (CD86 and CD80) and MHC Class I (MHC-I) molecules, as a measure of the immunological effects elicited on the target cells by our combination therapy. When exposed to NIR laser, cells treated with INAPs-PTT as well as ICG-PLGA-PTT attained similar maximum temperatures (~50 °C) and thermal doses (~1.6 log(CEM43) (Figure 5a). The specific temperatures and thermal doses administered were chosen to ensure that there were sufficient numbers of viable cells for the analysis and comparison of surface marker expression levels between groups. Cells treated with INAPs-PTT, which combined PTT and NextA therapy, expectedly exhibited the lowest viability of ~40% (Figure 5b), potentially from the slightly higher temperatures observed at minute 1 and 2 with INAPs-PTT (Figure 5a). In terms of the surface expression quantitated through the median fluorescence intensity (MFI), CD86, CD80, and MHC-I expression on SM1 tumor cells (Figure 5c–e) markedly increased after INAPs-PTT (red) and ICG-PLGA-PTT (blue) treatment compared to controls in vitro. The encapsulated NextA in the INAPs further enhanced the expression in the combined therapy with INAPs-PTT (red) in comparison to ICG-PLGA-PTT (blue). When expressed in terms of the percentage of cells expressing the aforementioned markers, INAPs-PTT and ICG-PLGA-PTT yielded 89.8% and 81.1% CD86 expression and 86.8% and 80.3% MHC-I expression (Appendix A), respectively. For CD80 expression, there was only a marginal increase in the percentage after INAPs-PTT compared to ICG-PLGA-PTT. In B16F10 cells, the INAPs-PTT and ICG-PLGA-PTT yielded 54.3% and 38.2% MHC-I expression, respectively (Appendix A). These results implicate that PTT greatly induces immune marker immunomodulation on tumor cells, when compared to NextA alone, and could be dependent on the thermal dose, although more studies would be needed to verify this phenomenon. Overall, these findings demonstrate that the simultaneous administration of photothermal and epigenetic therapy via INAPs-PTT induces increased expression of co-stimulatory molecules (CD86 and CD80) and MHC Class I expression relative to the control treatment, a measure of potentially improved immunological responses elicited by the combination therapy.

### 3.5. INAPs-PTT Slows Tumor Progression and Increases Median Survival in a Syngeneic Murine Model of Melanoma 

To evaluate the therapeutic efficacy of combining PTT with epigenetic therapy, we conducted a preliminary study in a syngeneic murine melanoma model. Specifically, mice bearing established (tumor volumes ~60 mm^3^) SM1 melanoma tumors were divided into the following five groups (*n* = 5 per group; Figure 6a): (1) Untreated—receiving no treatment; (2) NextA (i.p., 25 mg/kg)—receiving free NextA (epigenetic therapy) 6× per week through the entire study until the tumor exceeded 20 mm in any dimension; (3) INAPs (i.t., 50 mg/kg)—receiving i.t. encapsulated NextA (epigenetic therapy); (4) ICG-PLGA-PTT (i.t., 50 mg/kg + NIR laser for 10 min)—receiving PTT; (5) INAPs-PTT (i.t., 50 mg/kg +NIR laser for 10 min)—receiving both PTT and epigenetic therapy; and (6) INAPs-PTT (i.t., 50 mg/kg + NIR laser for 10 min) + 2 NextA-PLGA boosters (i.t. 50 mg/kg)—receiving both PTT and epigenetic therapy. Group 6 received NextA-PLGA boosters to maintain sufficient concentrations of NextA within the tumor microenvironment (TME) to complement the effects of PTT. Additionally, the dosing for free NextA in group 2 was based on the optimal doses previously determined to elicit antitumor activity [23,24,25]. The average final tumor temperature achieved during PTT measured by the thermal imaging camera was ~95°C and the corresponding thermal doses were ~16.8 log(CEM43) (Figure 6b). Groups receiving epigenetic therapy alone in free drug (NextA; yellow) or encapsulated drug (INAP; blue) form showed similar tumor growth curves to untreated tumor-bearing mice (Figure 6c). Importantly, nearly daily systemic administration of free NextA conferred no tumor progression benefit compared to untreated controls. By comparison, groups receiving PTT (ICG-PLGA-PTT; light blue) and PTT plus epigenetic therapy (INAPs-PTT + NextA-PLGA boosters; red) exhibited markedly slower tumor progression (Figure 6c). This sustained pressure of epigenetic therapy after PTT in the INAPs-PTT + NextA-PLGA boosters group also delayed the rate of tumor recurrence in treated mice (Table 1), which was not evident after only one INAPs injection without NextA-PLGA boosters as observed in the INAPs-PTT group. Treatment with ICG-PLGA-PTT and INAPs-PTT (for both the INAPs-PTT group and INAPs-PTT + NextA-PLGA booster group) effectively ablated the SM1 tumors on tumor-bearing mice, as the treated mice for each of these groups presented no tumor on day 1 post treatment (indicated by “no tumor” in Table 2). When the tumor progression in the INAPs-PTT-treated and INAPs-PTT + NextA-PLGA boosters-treated mice were compared to that of the ICG-PLGA-PTT-treated mice (Table 2), the sustained NextA pressure by the NextA-PLGA boosters caused a slower tumor progression compared to INAPs-PTT up until day 11. This was evident, for example, by day 8, where tumors in the INAPs-PTT + NextA-PLGA booster-treated group had an average tumor volume of only 19% that of the ICG-PLGA-PTT-treated group. In contrast, on the same day (day 8), this slower tumor growth was not evident in INAPs-PTT-treated mice as their tumor volumes were on average 156% that of ICG-PLGA-PTT-treated mice, indicating that INAPs-PTT tumors were on average 56% larger than ICG-PLGA-PTT tumors. At day 18 (9 days after the last NextA-PLGA booster), the benefit of controlling the tumor size with the sustained epigenetic therapy was no longer observed, as the tumor volume of the INAPs-PTT + NextA-PLGA booster group was 105% of that of the ICG-PLGA-PTT group, indicating that by day 18, tumors of the INAPs-PTT + NextA-PLGA booster group were on average 5% larger than in the ICG-PLGA-PTT group. Further, in terms of survival, there was no survival benefit observed in the tumor-bearing animals treated with free NextA or INAPs relative to untreated controls. However, all animals treated with PTT exhibited an increased median survival relative to untreated controls. The median survival was 18 days in the INAPs-PTT plus NextA-PLGA boosters group, 17 days in the ICG-PLGA-PTT, and 14 days in the untreated controls (Figure 6d), although there was no statistically significant differences in the long-term survival between the treatment groups. Overall, this preliminary study suggests that PTT combines with epigenetic therapy to slow tumor progression early after treatment and improves median survival in a syngeneic melanoma model. 

## 4. Discussion

In this study, we described a PLGA nanoparticle-based approach to combine photothermal and epigenetic therapy as a novel combination therapy for melanoma (Figure 1). Our nanoemulsion synthesis scheme resulted in the stable encapsulation of both ICG and NextA within 220 nm PLGA nanoparticles (INAPs; Figure 2). The consistent temporal size distributions of the INAPs (as measured by DLS) indicated stability of the INAPs in suspension. For translation purposes, high stability of therapeutic agents is critical for clinical applications and effective storage. However, upon administration, such PLGA-based nanoparticles would be expected to degrade overtime [34]. PLGA nanoparticle integrity can be affected by pH, with an acidic pH increasing the degradation rate and cargo release. Through their intratumoral localization, we expected that INAPs would not persist for long before the acidic tumor conditions initiate degradation of INAPs [3,35]. Current research on nanoparticle stability also demonstrate that the type of stabilizers used to coat and encapsulate drugs into nanoparticles could improve their stability properties that allow them to function as multifunctional carriers or enhance permeation through the skin to improve drug delivery [39,40]. Through UV-Vis-NIR spectrometry, we detected the encapsulation of ICG and/or NextA within the INAPs, ICG-PLGA, and NextA-PLGA. Although the amount of ICG loaded into the ICG-PLGA and INAPs were equivalent, as measured by their similar absorbance peaks (Figure 2d), the amount of NextA loaded into the INAPs and NextA-PLGA was significantly different (Figure 2e). Specifically, the amount of NextA loaded within the dual-agent INAPs was significantly higher than that loaded within the single-agent NAPs as evidenced by a higher NextA absorption band for INAPs compared to NAPs. One potential explanation is that with both ICG and NextA dissolved with PLGA in the organic phase, there is a higher solid-state drug–polymer solubility, which has been reported to increase drug loading since the entropy of the solution increases [41]. 

Through our analysis of INAPs to mediate PTT, we observed a difference in the temperatures and thermal doses achieved between the concentration-dependent and NIR laser power-dependent curves (Figure 3a–d). Based on these data, heating of INAPs is dependent on each batch of INAPs synthesized as there is a slight variability (40–50%) in the amount of ICG encapsulated, which results in small differences in the heating trajectories of the INAPs. However, regardless of this small variability between batches, the observed trends for increased heating with increasing concentration and laser power were retained. The laser exposure appeared to photobleach ICG and consequently diminish their PTT capabilities (Figure 3e). This bleaching of ICG after NIR laser exposure was evident in Figure 3a, specifically at 0.5 to 2.0 mg/mL, where the nanoparticle heating appeared to decrease after the initial heating. Thus, the amount of ICG encapsulated, the irradiation time, and the rate of photobleaching all impact the ability of INAPs to function as PTT agents, which provides further evidence for their use as a single administration of PTT. When melanoma cells were treated with INAPs-PTT in vitro, the encapsulated NextA had a minimal effect on viability, suggesting that the INAPs are releasing NextA at concentrations that have low cytotoxicity (Appendix A). 

The NextA encapsulated within PLGA nanoparticles (INAPs and NextA-PLGA) retained their ability to inhibit pan-HDAC activity (Figure 4a). This finding is consistent with earlier published reports where encapsulated agents in PLGA particles were shown to work similarly to or better than free agents (Appendix A) [4,7,8,9]. Critically, the activity of NextA was not diminished even in the presence of NIR laser activation, suggesting the compatibility of administering NextA with ICG-based PTT. The INAPs were able to exhibit concentration-dependent (Figure 4b) and temporal (Figure 4c) inhibition of HDAC activity, once again, both in the presence or absence of the NIR laser. The INAPs were able to inhibit HDAC6-specific activity as evidenced by the increased expression of Ac-α-tubulin in melanoma cells in vitro relative to controls (Figure 4d and Appendix A). The findings in Figure 3 and Figure 4 provide evidence for the use of PLGA nanoparticles to co-localize complementary therapies, such as PTT and epigenetic therapy, and is consistent with earlier studies using PLGA to co-localize therapies (Appendix A) [22,42].

Our studies demonstrated an improved expression of immunological markers expressed by melanoma cells induced by the combined photothermal-epigenetic therapy via the INAPs (Figure 5). Specifically, treatment with INAPs-PTT increased the expression of co-stimulatory markers CD80 and CD86 on melanoma cells in vitro. While tumor cells are not professional antigen-presenting cells, the expression of these co-stimulatory molecules on these cells helps T cells engage with tumor cells, facilitating the activation of primed effector T cells [43]. INAPs-PTT also increased MHC-I expression in melanoma cells in vitro. Since most cancers, including melanoma, evade immune detection by downregulating MHC-I, by inducing a higher MHC-I expression, we could potentially improve the tumor recognition by infiltrating CD8+ T cells through the MHC-I/TCR interaction [44]. While PTT alone (ICG-PLGA-PTT) increased CD80, CD86, and MHC-I expression, the addition of NextA further increased these expression levels on melanoma cells in vitro. It is important to note here that these studies were conducted at sub-lethal PTT conductions (~50 °C) to uncouple the cytotoxic effects of PTT from the immunomodulatory effects of PTT and NextA on tumor cells. Future studies would help validate if the increased expression of these immune markers (1) positively correlates with administered thermal temperature/doses for optimal PTT, and (2) if it could improve tumor-specific killing by activated T cells as a potential mechanism of action. 

In our in vivo studies in the syngeneic SM1 melanoma model, the combined therapy slowed early tumor growth and improved median survival (Figure 6). Sustained epigenetic therapy improved the response to PTT by delaying the timing of recurrence (Table 1), as observed in the early tumor growth curves (Figure 6) and by the tumor sizes for mice in the INAPs-PTT + NextA-PLGA booster-treated group compared to the ICG-PLGA-PTT-treated group (Table 2). Yet, it is important to note that the most prominent therapeutic efficacy was largely driven by PTT, as evidenced by nearly equal median survival for ICG-PLGA-PTT-treated mice compared with INAPs-PTT + NextA-PLGA boosters. The NextA-PLGA boosters post-INAPs-PTT appeared to improve the number of tumor-free days, which suggests that NextA acted on the tumor microenvironment to prevent tumor growth, but the modulation mediated by HDAC6 inhibition was not sufficient for maintaining long-lasting antitumor effects or improving long-term survival. Additional studies maintaining a sustained release of NextA would be needed to study the modulation in the TME driven by HDAC6 inhibition after INAPs-PTT. Further, a role of the drug in tumor priming prior to PTT cannot be discounted, which was not explicitly tested here.

In conclusion, by formulating a nanoparticle-based approach to combine photothermal and epigenetic therapy, we demonstrate the feasibility of combining disparate yet complementary therapies to treat melanoma. Characterization of the nanoparticles suggested that the INAPs could simultaneously administer both therapies in vitro, and our preliminary in vivo studies suggests the combined therapy elicits improved therapeutic efficacy early on during treatment. Future studies conducted to assess the immunomodulation elicited by each therapy alone and in combination will further describe the complementary antitumor effects of the combined photothermal-epigenetic therapy to maximize their therapeutic benefits for melanoma, and potentially other tumors.

## Figures and Tables

**Figure 1 nanomaterials-10-00161-f001:**
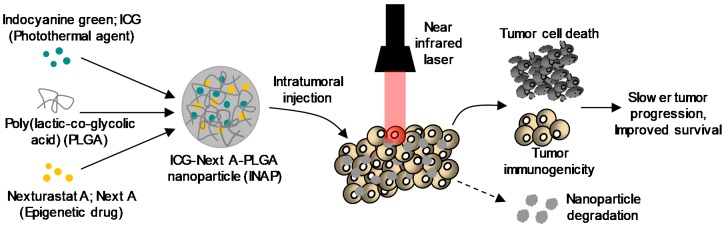
Hypothesized mechanism of action of our approach combining photothermal and epigenetic therapy for melanoma using poly (lactic-*co*-glycolic acid) (PLGA) nanoparticles. The photothermal agent, indocyanine green (ICG), and the epigenetic drug, Nexturastat A (NextA), were co-encapsulated within PLGA nanoparticles (ICG-Next A-PLGA; INAPs). The INAPs were administered to melanoma tumors in syngeneic murine models and activated with an 808 nm near infrared laser. ICG-based photothermal therapy along with concurrent epigenetic drug (NextA) release in the tumor microenvironment causes tumor cell death and increased immunogenicity, which we expected to slow tumor growth and improve survival in melanoma when compared to either therapy alone. Further, since PLGA is biodegradable, we expected our INAP platform to mitigate toxicity concerns associated with the long-term persistence of nanoparticles in vivo.

**Figure 2 nanomaterials-10-00161-f002:**
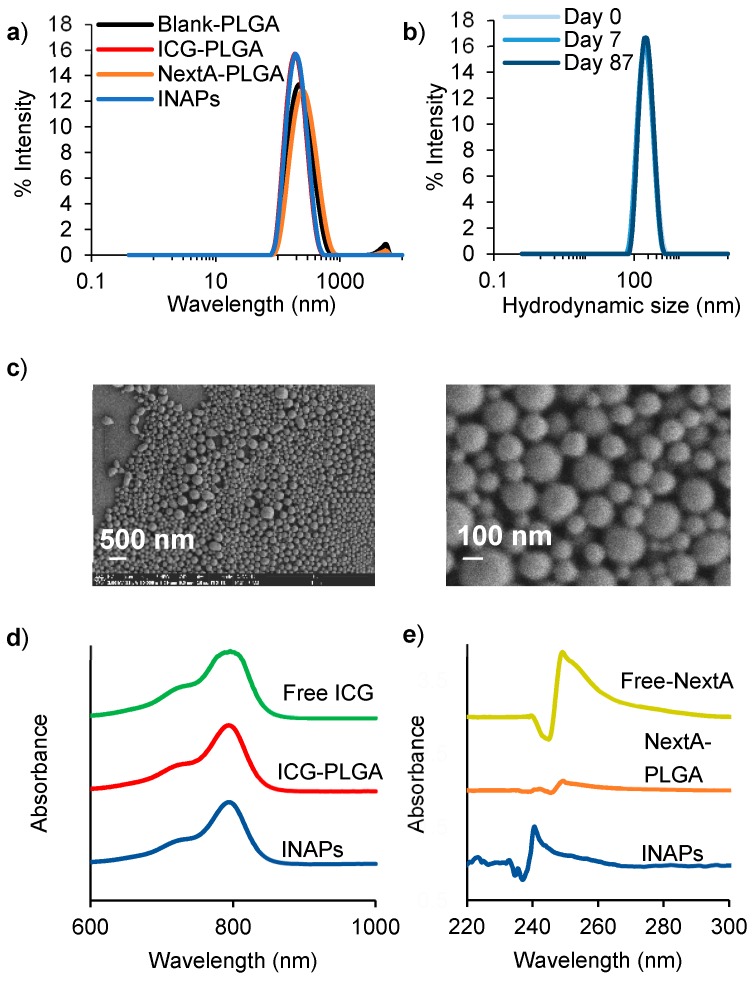
Characterization of INAPs demonstrate high stability and co-encapsulation of ICG and NextA. (**a**) Size distributions (hydrodynamic diameters) of INAPs, NextA-PLGA, ICG-PLGA, and Blank-PLGA measured by dynamic light scattering (DLS) ranged from 220 to 255 nm. (**b**) Multi-day stability of INAPs over 87 days as measured by DLS. (**c**) Scanning electron microscope images of INAPs at high magnification (50,000×, curr 13 pA), zoomed out (left) and zoomed in (right). UV-Vis-NIR spectrometry measured the absorption peaks of (**d**) ICG (free agent; green) at 780 nm and (**e**) NextA (free agent; gold) at 250 nm to determine the presence of ICG in ICG-PLGA (red) and INAPs (blue), and NextA in NextA-PLGA (orange) and INAPs (blue). INAPs, ICG-PLGA, and NextA-PLGA spectra were normalized to Blank-PLGA spectra.

**Figure 3 nanomaterials-10-00161-f003:**
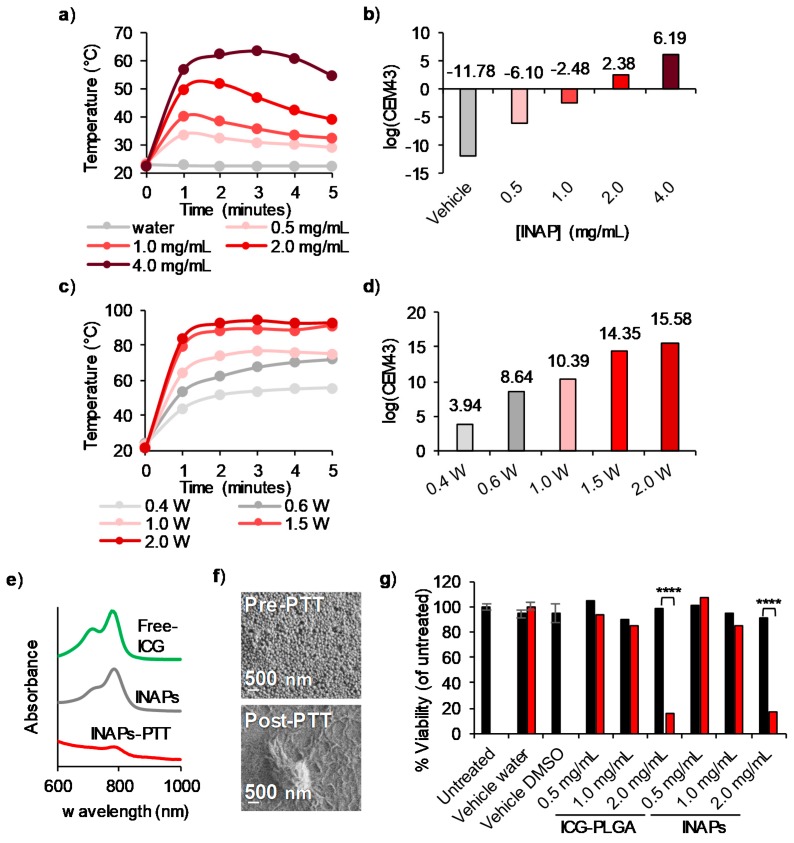
The photothermal heating of INAPs decreases melanoma cell viability in vitro. (**a**) Time–temperature heating curves at 1-min intervals and (**b**) Thermal doses expressed in log(CEM43) of varying concentrations of INAPs (0.5–4 mg/mL) exposed to a 0.8 W NIR laser for 5 min demonstrate concentration-dependent heating. (**c**) Time–temperature heating curves at 1-min intervals and (**d**) thermal doses expressed in log(CEM43) of 4 mg/mL INAPs exposed to varying NIR laser powers (0.4–2 W) for 5 min demonstrate laser power-dependent heating. (**e**) UV-Vis-NIR spectroscopy of free ICG (green), INAPs before (gray), and after (red) exposure to 0.8 W NIR laser for 5 min show lower ICG presence. (**f**) SEM images of INAPs show changes in INAP morphology before (Pre-PTT, mag. 50,000×, curr 25 pA) and after (Post-PTT, mag. 50,000×, curr 13 pA) exposure to a 0.8 W NIR laser for 5 min. (**g**) SM1 melanoma cells were suspended in PBS and treated with INAPs or ICG-PLGA at varying concentrations or controls in the presence (red) or absence (black) of the laser. Cells were subsequently re-plated in media and incubated at 37 °C for 24 h, after which the Cell Titer-Glo ATP assay was performed and determined a decrease of the SM1 cell viability in the presence of the laser.

**Figure 4 nanomaterials-10-00161-f004:**
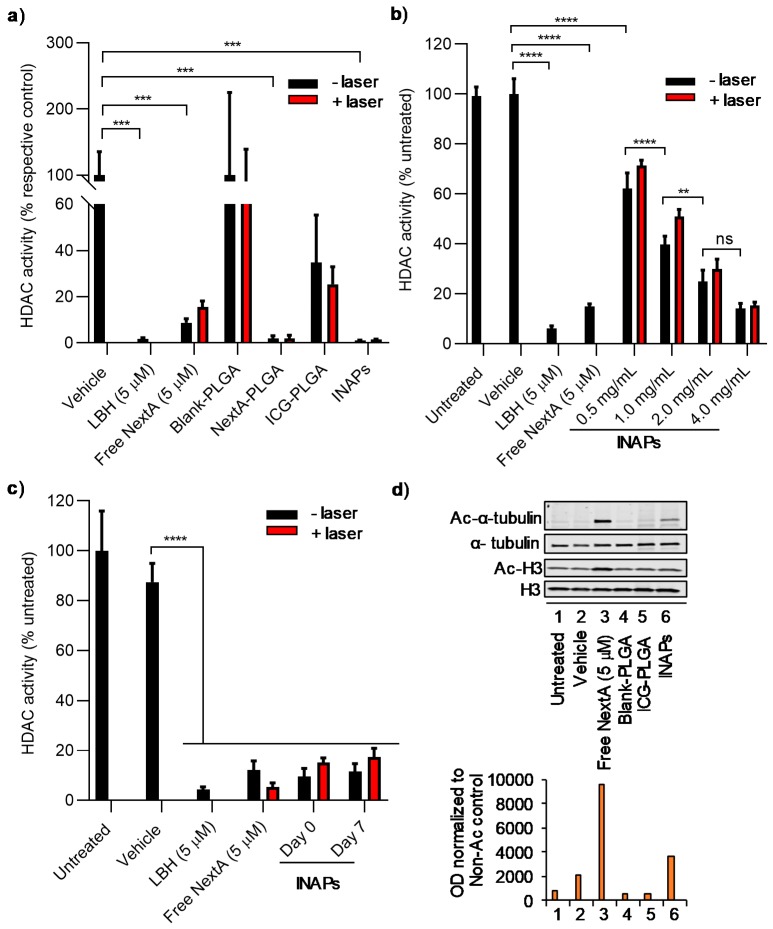
Inhibition of HDAC activity in melanoma cells in vitro by NextA encapsulated within INAPs. (**a**) HDAC activity of SM1 melanoma cells in vitro measured by a luminescence reporter assay when treated with INAPs, ICG-PLGA, NextA-PLGA, Blank-PLGA, free NextA, a positive control (panobinostat; LBH), and vehicle in the absence (black) and presence (red) of the NIR laser at 0.8 W for 5 min. HDAC activity decreased after treating with INAPs, NextA-PLGA, free NextA, and LBH. *** *p*-value < 0.001. (**b**) HDAC activity (luminescence) of B16F10 melanoma cells in vitro treated with varying concentrations of INAPs (0.5–4 mg/mL) and controls in the absence (black) and presence (red) of the NIR laser at 0.8 W for 5 min showed concentration-dependent HDAC inhibition. ns (not signficiant), ** *p*-value < 0.01, **** *p*-value < 0.0001. (**c**) HDAC activity (luminescence) of B16F10 melanoma cells in vitro treated with freshly synthesized (Day 0) or short-term stored INAPs (Day 7) and controls in the absence (black) and presence (red) of the NIR laser at 0.8 W for 5 min demonstrated functionally active NextA in INAPs after storage. **** *p*-value < 0.0001. (**d**) Expression levels of Ac-α-tubulin, α-tubulin, and acetylated histone 3 (Ac-H3) and histone 3 (H3) in B16F10 melanoma cells treated with INAPs and controls in vitro demonstrated increased Ac-α-tubulin expression after NextA and INAPs treatment. Top panel: Western blot image of the various treatment groups, bottom panel: Image analysis of the expression levels in the top panel (using Image J, National Institutes of Health, Bethesda, MD, USA).

**Figure 5 nanomaterials-10-00161-f005:**
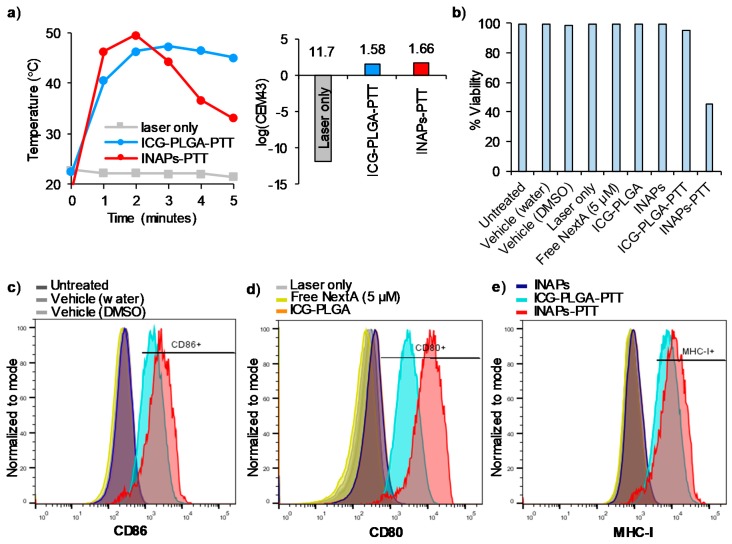
Increased cell surface expression of co-stimulatory molecules and MHC Class I on melanoma cells in vitro after INAPs-PTT. (**a**) Time–temperature heating curves at 1-min intervals and thermal doses expressed in terms of log (CEM43) of SM1 melanoma cells treated with INAPs-PTT and ICG-PLGA-PTT using a 0.4 to 0.6 W NIR laser for 5 min showed heating to approximately 50 °C with a log (CEM43) value of ~1.6. (**b**) Viability of SM1 melanoma cells treated with INAPs-PTT, ICG-PLGA-PTT, and other control treatment groups showed 40% or higher viability after treatment, as measured by flow cytometry. Cell surface expression levels of co-stimulatory molecules (**c**) CD86, (**d**) CD80, and (**e**) MHC-I increased after INAPs-PTT and ICG-PLGA-PTT but not with other control treatment groups, as measured by flow cytometry (panels c–e x-axis units: logarithmic fluorophore expression levels, y-axis units: acquired events normalized to mode).

**Figure 6 nanomaterials-10-00161-f006:**
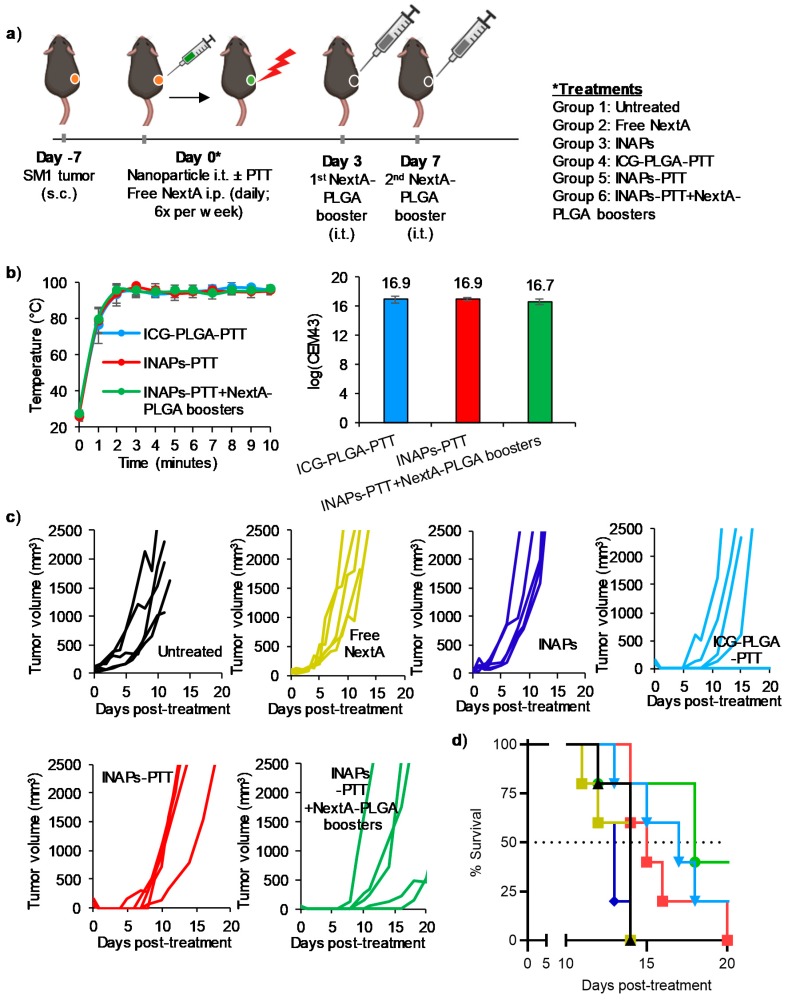
INAPs-PTT slows tumor progression and increases median survival in local melanoma-bearing mice. SM1 melanoma cells were inoculated on the right flank of C57BL/6 mice. (**a**) Schematic for the combination therapy in a primary tumor model. (**b**) Temperatures for PTT groups were measured every minute by a thermal camera and maintained at approximately 95 °C, with the corresponding log(CEM43) ~16.9. (**c**) Individual tumor growth curves (*n* = 5/group) and (**d**) the overall Kaplan–Meier survival plots demonstrate short-term delay of tumor recurrence and increased median survival with sustained epigenetic therapy after PTT (INAPs-PTT + NextA-PLGA boosters).

**Table 1 nanomaterials-10-00161-t001:** Comparison of the number of tumor-bearing mice several days after ICG-PLGA-PTT, INAPs-PTT, and INAPs-PTT + NextA-PLGA booster treatments.

Number of Tumor-Bearing Mice			
Days Post-Treatment	ICG-PLGA-PTT	INAPs-PTT	INAPs-PTT + NextA-PLGA Boosters
Pre-Treatment	5	5	5
1	0	0	0
3	0	0	0
7	2	1	0
8	2	4	1
11	4	5	4

**Table 2 nanomaterials-10-00161-t002:** Quantitative comparison of tumor sizes in mice several days after INAPs-PTT and INAPs-PTT + NextA-PLGA booster treatments, relative to ICG-PLGA-PTT.

Tumor Sizes Compared to ICG-PLGA-PTT (Expressed as % of ICG-PLGA-PTT Tumor on a Particular Day)			
Days Post-Treatment	ICG-PLGA-PTT	INAPs-PTT ^1^	INAPs-PTT + NextA-PLGA Boosters ^2^
1	no tumor	no tumor	no tumor
7	100%	53%	0
8	100%	156%	19%
11	100%	169%	27%
18	100%	162%	105%

^1^ Last day for NextA administration (via INAPs) was day 0; ^2^ Last day for NextA administration (via NextA-PLGA booster) was day 7.

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
