# Peer review of "Indocyanine Green-Nexturastat A-PLGA Nanoparticles Combine Photothermal and Epigenetic Therapy for Melanoma"

_nanomaterials, 2020, doi:10.3390/nano10010161_

Round 1

Reviewer 1 Report

The manuscript shows a detailed in-vivo and in-vitro study using PLGA particles that encapsulates FDA approved photothermal and epigenetic agents for melanoma therapy.

I recommend to accept the paper after minor revisions I expect the article to make a significant contribution to the literature.

1) The paper includes many abbreviations and sometimes it overwhelms the reader. Simplify the number or make it simple to quickly understand which therapy was used. Use longer and easy nomenclature:

free ICG, ICG-PLGA, NextA-PLGA, ICG-Next-PLGA, free ICG-NextA, free NextA, PLGA.

NPs and NAPs can be misinterpreted.

2) Include a final table in the discussion section to follow better the comparison with previous papers.

3) Include error bars in the graphs/data and units for all axis, example Figure 5 c-e.

4) Explain >100% in no tumor for table 2.

5) Include some further discussion about the % drug/agent that is loaded/unloaded from the PLGAs versus free-drug to better understand the behavior in the tumor.

Reviewer 2 Report

The paper by Ledezma et al. suggests the use of a combined approach to the in situ PPT treatment of melanoma using PLGA nanoparticles loaded with both a PPT agent and an epigenetic drug. Nanoparticles production and characterization and in vitro and in vivo experiments are well designed and results are sounded. However, in order to understand the value of a such novel approach,  I would have also used a lethal PPT conduction as a positive control.

Reviewer 3 Report

The comments are attached below as a pdf file.

Round 2

Reviewer 3 Report

I accept the changes provided and recommend the current version of the manuscript for publication in Nanomaterials.